# Voluntary medical male circumcision in selected provinces in South Africa: Outcomes from a programmatic setting

Khumbulani Moyo[1], Nelson Igaba[1], Constance Wose Kinge[1,2], Charles Chasela[1,2]*, Motshana Phohole[1], Skye Grove[1], Caroline Makura[1], Latisani Mudau[1], Dirk Taljaard[3], Dino Rech[3], Arthi Ramkissoon[4†], Catherine Searle[4], Pappie Majuba[1], Ian Sanne[1,5]

1 Right to Care, Johannesburg, South Africa, 2 Department of Epidemiology and Biostatistics, School of Public Health, Faculty of Health Sciences, University of the Witwatersrand, Johannesburg, South Africa, 3 Centre for HIV and AIDS Prevention Studies (CHAPS), Johannesburg, South Africa, 4 Maternal Adolescent and Child Health (MatCH), Durban, South Africa, 5 Clinical HIV Research Unit, Department of Clinical Medicine, Faculty of Health Sciences, University of the Witwatersrand, Johannesburg, South Africa

† Deceased.

* charles.chasela@wits.ac.za

## Abstract

### Introduction

Voluntary medical male circumcision (VMMC) remains an effective biomedical intervention for HIV prevention in high HIV prevalence countries. In South Africa, United States Agency for International Development VMMC partners provide technical assistance to the Department of Health, at national and provincial levels in support of the establishment of VMMC sites as well as in providing direct VMMC services at site level since April 2012. We describe the outcomes of the Right to Care (RTC) VMMC program implemented in South Africa from 2012 to 2017.

### Methods

This retrospective study was undertaken at RTC supported facilities across six provinces. Young males aged ≥10 years who presented at these facilities from 1 July 2012 to 31 September 2017 were included. Outcomes were VMMC uptake, HIV testing uptake and rate of adverse events (AEs). Using a de-identified observational database of these clients, summary statistics of the demographic characteristics and outcomes were calculated.

### Results

There were a total 1,001,226 attendees of which 998,213 (99.7%) were offered VMMC and had a median age of 15 years (IQR = 12–23 years). Of those offered VMMC, 99.6% (994,293) consented, 96.7% (965,370) were circumcised and the majority (46.3%) were from Gauteng province. HIV testing uptake was 71% with a refusal rate of 15%. Of the newly diagnosed HIV positives, 64% (6,371 / 9,972) referrals were made. The rate of AEs, defined as bleeding, infection, and insufficient skin removal) declined from 3.26% in 2012 to

**Data Availability Statement:** Data cannot be shared publicly but available upon request from the Institution as per organization data governance policy. Requests can be directed through the data

governance unit contact person Mr Tapiwa Mandizwidza Email: apiwa. Mandizvidza@righttocare.org.

**Funding:** This study was funded by USAID through a grant awarded to KM (AID-674-Q-13-00002). The funders had no role in study design, data collection and analysis, decision to publish, or preparation of the manuscript.

**Competing interests:** The authors have declared that no competing interests exist.

1.17% in 2017. There was a reduction in infection-related AEs from 2,448 of the 2,602 adverse events (94.08%) in 2012 to 129 of the 2,069 adverse events (6.23%) in 2017.

## Conclusion

There was a high VMMC uptake with a decline in AEs over time. Adolescent men contributed the most to the circumcised population, an indication that the young population accesses medical circumcision more. VMMC programs need to implement innovative demand creation strategies to encourage older males (20–34 years) at higher risk of HIV acquisition to get circumcised for immediate impact in reduction of HIV incidence. HIV prevalence in the total population increased with increasing age, notably in clients above 25 years.

## Introduction

Voluntary medical male circumcision (VMMC) remains one of the key interventions for HIV prevention in countries with high HIV prevalence [1]. Further to the three randomized controlled trials (RCTs) where medical male circumcision (MMC) resulted in a 60% reduction in the risk of female-to-male HIV transmission [2–4], a recent systematic review and meta-analysis demonstrated reduction in risk of HIV infection in the post-RTC follow-up, in community-based and in circumcision scale-up studies [1]. From 2008 to 2019, nearly 27 million adolescent and adult men (≥10 years) had been circumcised and an estimated 340 000 new infections averted in 15 VMMC priority countries, including 260 000 infections among males and 75 000 among females (due to reduced secondary transmission from males [5]. Furthermore, VMMC programs are considered highly effective in reducing both HIV incidence and the cost of HIV prevention especially when targeting young males (20–35 years old) who are at higher risk of HIV acquisition [6]. Male circumcision only offers partial protection against HIV transmission, hence, MMC must be integrated with a comprehensive HIV prevention strategy, which includes treatment for sexually transmitted infections, HIV testing and counselling and promotion of safe sex practices [7] and provision of pre-exposure prophylaxis (PrEP).

Traditional circumcision, also viewed as a "rite of passage into manhood" is a common practice among the Xhosa, Sotho, Pedi, Venda, and Ndebele cultural groups of South Africa [8]. This "traditional rite" is usually performed in the months of June, July, November, and December, commonly referred to as the circumcision season. The procedure is performed by traditional leaders without adequate infection control measures in place. As a result, traditional circumcision is often associated with multiple surgical complications and increased risk of injury and even death among young men and boys [9].

In 2012, the United States Agency for International Development (USAID) awarded Right to Care (RTC) a contract to implement a VMMC program across the provinces of Gauteng, KwaZulu-Natal (KZN), Free State, Limpopo, Mpumalanga, and North West in South Africa. RTC implemented the VMMC program in partnership with the Centre for HIV/AIDS Prevention Studies (CHAPS), ANOVA Health Institute and Maternal Adolescent and Child Health (MatCH). In this paper, we report the outcomes of the VMMC program implemented across the six provinces from 2012 to 2017.

## Materials and methods

### Study design

This was retrospective data analysis using routinely collected program data by the South African Department of Health USAID-funded VMMC program for the period 1 July 2012 to 30 September 2017.

### Study setting

The VMMC program operated in 149 sites across selected districts in Gauteng, Free State, KwaZulu-Natal, Limpopo, Mpumalanga and North West Provinces. The districts include City of Johannesburg, City of Tshwane, Ekurhuleni, Sedibeng and West Rand for Gauteng; Fezile Dabi and Lejweleputswa for Free State; eThekwini, Ugu, uMgungundlovu, uMkhanyakude and Zululand for KwaZulu-Natal; Capricorn, Mopani and Vhembe for Limpopo; Ehlanzeni and Gert Sibande for Mpumalanga; and Bojanala Platinum for North West. The sites in Limpopo, Mpumalanga, Free State and North West Provinces cater mainly for the rural population while those in the Gauteng and Kwa-Zulu Natal Provinces cater for the urban population. Selection of sites was determined by the funder (USAID) and the South Africa Department of Health at national and provincial levels.

### Participants

All males aged ≥10 years were eligible for circumcision. This age group is in line with the South African MMC National Guidelines [10]. Written informed consent for MMC was required and given independently by all males aged ≥18 years. Boys aged 16–17 years provided assent to the circumcision procedure after being given information with their parent or legal guardian giving written informed consent. All boys aged 10–15 years required parental/guardian written informed consent to undergo MMC. They also gave assent, and the parent/guardian was required to be present on the day of the circumcision.

### Recruitment of participants

Demand creation activities were undertaken through campaigns at community level, messaging through social media, radio, and other forms of media. All those willing were booked and referred to a facility for further screening, consenting and circumcision. At the facility, clients were recruited through walk- ins and referrals from other service points.

### Circumcision procedures

Once a client was recruited at the facility, the client was registered and provided with general group education. Following group education, individual counselling and HIV testing was offered, and subsequently MMC to all eligible clients (**Fig 1**).

Pre-operative history taking, physical examination and circumcision procedures were carried out among clients who accepted and consented. VMMC was deferred for new HIV positive clients when available CD4 count was <350 cells/µL but referred for antiretroviral therapy (ART) initiation. For known HIV positive clients, an assessment of the client's adherence to ART and retention in HIV care was conducted. The client was referred for care if he was lost to follow up. Eligibility for circumcision for such clients was based on recent CD4 count ≥350 cells/µL and/or viral load (VL) of Lower Than Detectable Limits (LTDL) with CD4 and VL results not more than six months old. Circumcision was deferred if VL was ≥1000 copies/ml regardless of CD4 value.

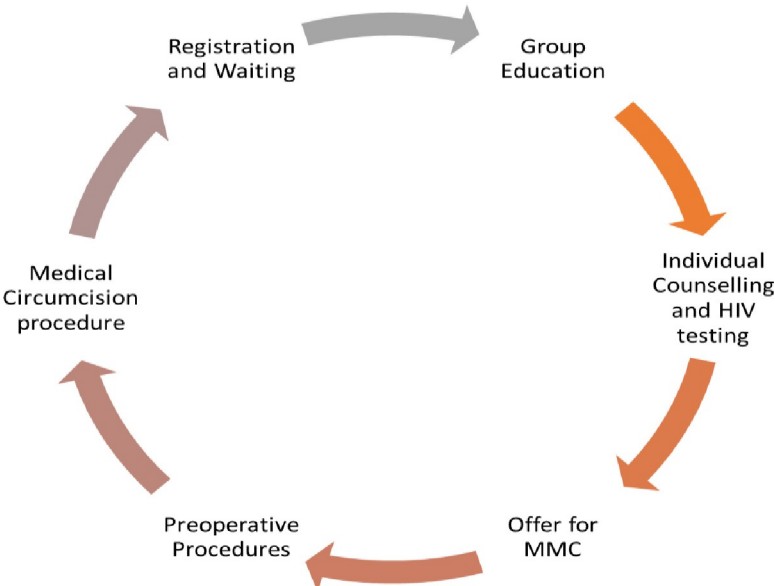

**Fig 1. Patient flow for VMMC at site of operation.**

## VMMC site team composition

The team composition was built to ensure optimization of volumes and efficiency in the delivery of services, a model called MOVE "Model for Optimizing Volume and Efficiency" [11]. The MOVE team comprised of a Surgeon (Clinical Associate or Medical Officer), Professional Nurse (PN), two enrolled nurses, two counsellors, administration clerk, a cleaner and a driver.

## Post-operative follow ups

Following the MMC procedure, follow up visits were scheduled to assess for wound healing and any signs of AEs. The first assessment occurred immediately after the procedure to check for signs of bleeding or any event and if there was no problem, the client was discharged from the facility. The subsequent follow-up visits were scheduled on 2, 7- and 42-days post-circumcision.

## Data management and analysis

Client data captured by RTC data capturers into RightMax (a cloud-based database for financial and programmatic reporting and monitoring purposes) for the period of 1 July 2012 to 30 September 2017 was retrieved and exported into Microsoft Excel. Client name and surname, date of birth, and ID or passport number as well as phone numbers were deleted. The de-identified data was then imported into Stata version 15 for further management and analysis. Non-eligible clients were excluded from the analysis where necessary. Proportions were calculated to describe the population characteristics as well as the outcomes and consenting, age categories, circumcised, HIV testing and status of all attendees including females.

## Ethical considerations

Informed assent/consent was obtained from clients or their parents/guardians as part of program requirements. Ethics clearance was obtained from the University of the Witwatersrand, Johannesburg Human Research Ethics Committee (#M150823).

## Results

### Participants

Out of 1,001,266 attendees, 99.7% (998,213) were offered VMMC and of these,99.6% (994,293) accepted and consented for VMMC. Of those that consented, 97.2 (965,370) were circumcised (**Fig 2**). The median age of the participants was 15 years (IQR = 12–23 years) and 65.7% were between ages 10 and 19 years (n = 655,490). Most clients were from Gauteng (46.3%), followed by Mpumalanga Province (20%) and most circumcisions were done between 2013 and 2017 (**Table 1**).

Clients who consented but did not undergo circumcision were those that either tested HIV positive on the day of circumcision and had CD4 <350 cell/uL or had one of the contra-indications as listed in the South African MMC guidelines [10]. Those that tested HIV positive were referred for ART initiation and those with either infection, uncontrolled chronic illnesses or penile anatomical abnormalities were referred for further management as per the South African National Department of Health (NDoH) referral guidelines [10].

### HIV testing and linkage to care

A total of 1,001,088 were offered an HIV test and 71.1% were tested, 15% declined the test and the rest had a known status. These included men and women who presented at the facility. Out of those newly diagnosed, referrals were undertaken in 86.6% (**Table 1**). The HIV prevalence increased with age with the highest among those ≥30 years (**Table 2**).

### Adverse events (AEs)

There were 10, 608 reported AEs between 2012 and 2017 with the highest numbers reported among clients aged 15–19 years (n = 3, 048; 28.73%) (**Fig 3**). Out of these, 58.81% (n = 6, 239)

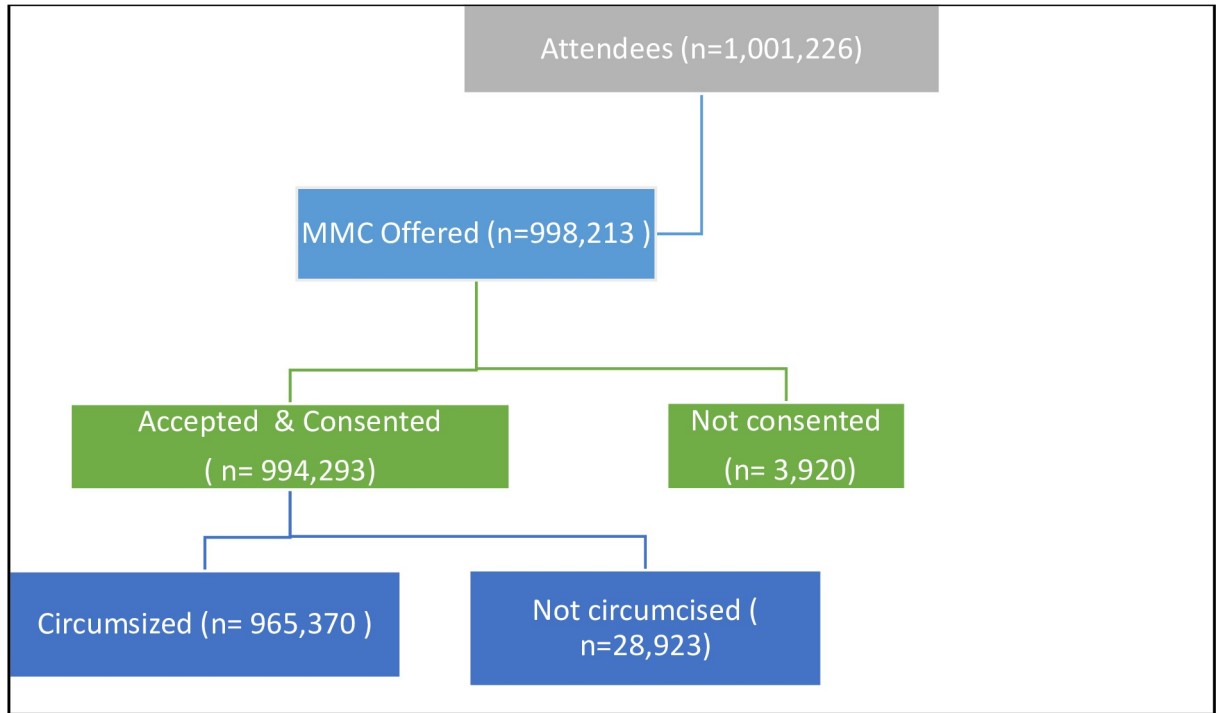

**Fig 2. Flow of participants for the VMMC program.**

**Table 1. Description of participants characteristics.**

| Characteristics | N | % |
|---|---|---|
| **Age (median, IQR)** | 15 | 12–23 |
| **Age Categories (%)** | | |
| 10–14 | 449,683 | 45.05 |
| 15–19 | 205,807 | 20.62 |
| 20–24 | 121,723 | 12.19 |
| 25–29 | 89,184 | 8.93 |
| 30–34 | 59,053 | 5.92 |
| 35–39 | 34,600 | 3.47 |
| 40 and Older | 38,163 | 3.82 |
| **Province (%)** | | |
| Free State | 41,238 | 4.13 |
| Gauteng | 462,128 | 46.30 |
| KwaZulu-Natal | 145,801 | 14.61 |
| Limpopo | 94,875 | 9.50 |
| Mpumalanga | 199,380 | 19.97 |
| North West | 54,791 | 5.49 |
| **Year (%)** | | |
| 2012 | 81,941 | 8.21 |
| 2013 | 139,931 | 14.02 |
| 2014 | 200,515 | 20.09 |
| 2015 | 206,163 | 20.65 |
| 2016 | 188,268 | 18.86 |
| 2017 | 181,395 | 18.17 |
| *****HIV testing offered (%)** | | |
| Tested | 711,882 | 71.11 |
| Not tested (Known status) | 139,225 | 13.91 |
| Declined tested | 149,981 | 14.98 |
| ***** **HIV Status (%)** | | |
| Negative | 692,880 | 69.21 |
| Known negative | 115,638 | 11.55 |
| Positive | 18,673 | 1.87 |
| Known positive | 23,596 | 2.36 |
| Unknown | 150,301 | 15 |
| ***** **Newly Diagnosed HIV (%)** | | |
| Not referred to treatment | 2,503 | 13.40 |
| Referred to treatment | 16,170 | 86.60 |

***** The number of HIV testing offered included female attendees but not males less than 10 years.

were infection-related followed by 8.30% (n = 880) due to bleeding, 0.92% (n = 98) were due to insufficient skin removal and 31.97% (n = 3, 391) of the AEs had missing AE type. The rate of AEs reduced over time with the highest incidence of 71.66%, (n = 2, 602) in the year 2012 and 1.97% (n = 2, 004) in 2013 (**Table 3**). There was a reduction in infection-related AEs from 94.08% (2, 448 of 2602) in 2012 to 6.23% (129 of 2069) in 2017 of the total AEs reported each year over the five-year period (**Table 4**). Among all identified AEs, 3.99% (n = 423) were related to use of Prepex device, 32.70% (n = 3, 469) were surgical related. However, 31.97% (n = 3,391) had missing technique of circumcision (**Table 4**).

**Table 2. HIV Status by age groups.**

| Age | HIV Status, n (%) | | | | | |
|---|---|---|---|---|---|---|
| | Negative | Positive | Known Negative | Known Positive | Unknown | Total |
| 10–14 | 283, 680 (63.06) | 2, 391 (0.53) | 48, 990 (10.89) | 3, 428 (0.76) | 111, 338 (25) | 449, 827 (100) |
| 15–19 | 157, 737 (76.51) | 1, 241 (0.60) | 32, 144 (15.59) | 1, 414 (0.69) | 13, 631 (7) | 206, 167 (100) |
| 20–24 | 95, 913 (78.39) | 1, 841 (1.50) | 15, 752 (12.87) | 1, 1141 (0.93) | 7, 698 (6) | 122, 347 (100) |
| 25–29 | 67, 195 (74.88) | 3, 725 (4.15) | 8, 778 (9.84) | 3, 217 (3.58) | 6, 823 (8) | 89, 738 (100) |
| 30–34 | 41, 117 (69.20) | 3, 939 (6.63) | 4, 696 (7.95) | 4, 692 (7.90) | 4, 976 (8) | 59, 420 (100) |
| 35–39 | 22, 688 (65.10) | 2, 668 (7.66) | 2, 396 (6.92) | 4, 222 (12.11) | 2, 876 (8) | 34, 850 (100) |
| 40 and older | 24, 550 (63.37) | 2, 868 (7.40) | 2, 882 (7.55) | 5, 4812 (14.15) | 2, 957 (8) | 38, 739 (100) |
| Total | **692, 880 (69.21)** | **18, 673 (1.87)** | **115, 638 (11.55)** | **23, 596 (2.36)** | **150, 301 (15)** | **998, 214 (100)** |

## Discussion

In this paper, we report the outcomes and trends for a USAID-funded VMMC program over a five–year period (2012–17) across six provinces in South Africa. A majority of mobilized VMMC eligible attendees were between the ages 10 to 24 years and were offered MMC. This is indicative of a young population seeking VMMC services. A review of VMMC programs in 2018 in fifteen eastern and southern African countries including South Africa also found that about 84% of clients in twelve of the fifteen countries were young men 10 to 29 years old with majority being 10 to 14 years old [5, 12]. In addition, our study shows that the program reached fewer young men aged 25–34 years, who are sexually active and at risk for HIV

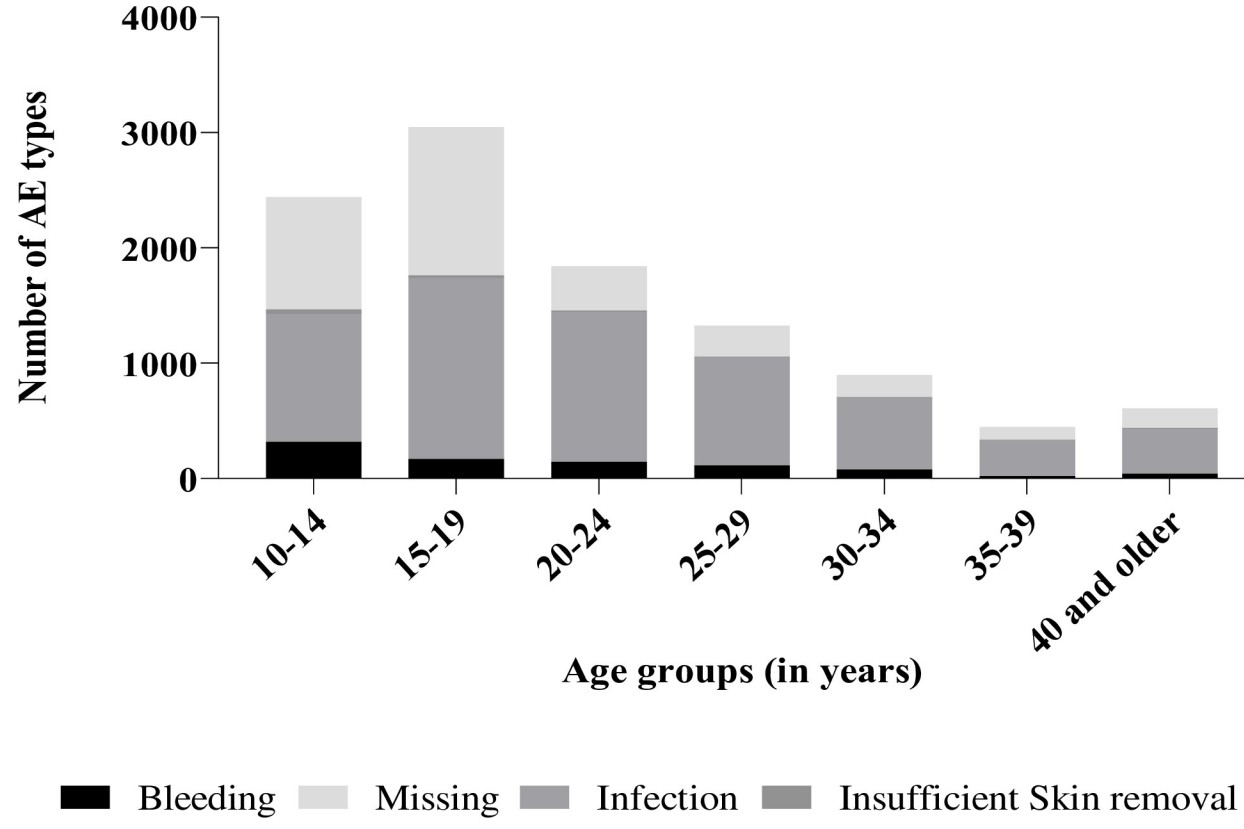

**Fig 3. Number of AEs identified by type per age group.**

**Table 3. Adverse events rates.**

| Year | Timing | | Severity | | | Total AEs | VMMCs Done | Follow-up | AE Rate (%) |
|---|---|---|---|---|---|---|---|---|---|
| | Intra-operative | Post-operative | Mild | Moderate | Severe | | | | |
| **2012** | 11 | 2, 591 | 0 | 1, 968 | 634 | 2, 602 | 79, 820 | 3, 631 | 71.66 |
| **2013** | 5 | 1, 999 | 0 | 1, 831 | 173 | 2, 004 | 136, 163 | 16, 745 | 11.97 |
| **2014** | 35 | 1, 382 | 0 | 1, 385 | 32 | 1, 417 | 194, 359 | 55, 614 | 2.55 |
| **2015** | 15 | 471 | 0 | 448 | 38 | 486 | 198, 587 | 81, 201 | 0.60 |
| **2016** | 1, 630 | 400 | 1, 488 | 452 | 90 | 2, 030 | 179, 904 | 106, 661 | 1.90 |
| **2017** | 1, 930 | 139 | 1, 528 | 391 | 150 | 2, 069 | 176, 537 | 72, 411 | 2.86 |
| **Total** | **3, 626** | **6, 982** | **3, 016** | **6, 475** | **1, 117** | **10, 608** | **965, 370** | **336, 263** | **3.15** |

acquisition. To achieve immediate impact in reduction of HIV incidence in these districts, the VMMC program needs to target males 20–34 years old. This is in line with WHO and NDoH guidance on maximizing impact of VMMC in HIV prevention [10, 13] by increasing uptake of VMMC services among adult men and especially those who may be at higher risk of HIV infection, such as partners of sex workers, men in sero-discordant relationships and men attending STI clinics [13]. In order to reach this age group, VMMC programs need to implement innovative demand creation strategies that may include: 1. Scale-up of sector specific approaches for work-based VMMC services like in mines, farms, military residences and other places of work; 2. Introduction of incentives and loss of income vouchers for older males working [13]; 3. Provision of male friendly services that include school based campaigns, extended hours of services and access to services over weekends, 4. Availing outreach and mobile services to sports grounds and higher institutions of learning; 5. Provision of one stop centers with a complete package of men's health services like sexual and reproductive health services, PrEP, and family planning; and 6. Policy adjustment on issues that affect the health of adolescent boys and men while seeking health care services [13] Out of the six provinces, most circumcisions were undertaken in Gauteng Province with increasing number of circumcisions over time. The Thembisa modelling 2.0 estimates that most provinces in South Africa have not reached the VMMC saturation mark of circumcising 80% of males 15–49 years old. Limpopo province is the only province with an estimate circumcision saturation of more than 80% and this could be attributed to their practice of traditional male circumcision of young males as early as 10 years of age [14]. The South African VMMC program will need to implement innovative ways of scaling up VMMC services to provinces with high HIV prevalence and low circumcision saturation like KwaZulu Natal and Mpumalanga. The COVID-19 pandemic has impacted negatively on the VMMC program in South Africa through suspension of the program for eight months and repurposing of VMMC facilities and staff for COVID-19

**Table 4. Types of Adverse events from 2012–2017.**

| Year | AE Type, n (%) | | | | |
|---|---|---|---|---|---|
| | Bleeding related | Infection related | Insufficient Skin Removal | Missing | Total |
| 2012 | 134 (5.15) | 2, 448 (94.08) | 20 (0.77) | 0 (0.00) | 2, 602 (100) |
| 2013 | 244 (12.18) | 1, 741 (86.88) | 19 (0.95) | 0 (0.00) | 2, 004 (100) |
| 2014 | 162 (11.43) | 1, 242 (87.65) | 13 (0.92) | 0 (0.00) | 1, 417 (100) |
| 2015 | 86 (17.70) | 373 (76.75) | 27 (5.56) | 0 (0.00) | 486 (100) |
| 2016 | 111 (5.47) | 306 (15.07) | 16 (0.79) | 1, 597 (78.67) | 2, 030 (100) |
| 2017 | 143 (6.91) | 129 (6.23) | 3 (0.14) | 1, 794 (86.71) | 2, 069 (100) |
| Total | 880 (8.30) | 6, 239 (58.81) | 98 (0.92) | 3, 391(31.97) | 10, 608 (100) |

management. Extra efforts and commitment will be required to scale-up VMMC in South Africa to reach the set targets.

There was a high acceptance of HIV testing among clients who came for VMMC services, with 14% arriving with known status results and 15% declining testing. HIV prevalence was about 2% among the newly diagnosed and 3% among participants who were known positives. HIV prevalence was high among those above 25 years and positivity increased with age, with a positivity rate of about 8% among those newly diagnosed and between the ages 35–39 and 12–14% among those above 40 years of age. The 3% among participants who were known positives was lower compared to general HIV prevalence in South Africa which is 13.7% [15]. However, the rate of positivity was high among the newly tested in the ages above 35 years. The relatively high HIV testing refusal rate of 15% is in line with findings from another study where program data from fourteen southern and east African countries indicated unknown HIV status among participants ranging from 0% to 50% [12]. The higher HIV positivity rate in older men aged >20 years is indicative of the need to focus efforts for VMMC and other HIV prevention modalities amongst this age group to create an immediate impact on reduction of HIV incidence in these communities. The VMMC program targets young healthy HIV negative males to confer to them the benefit of reduced transmission of HIV from an HIV positive female partner by 60%. To increase demand and ensure effective linkage to care and treatment of newly tested HIV positive men, VMMC programs need to collaborate with HIV care and treatment programs to implement a two-way referral system that will facilitate referral of HIV negative males for VMMC services and HIV positive men for HIV care and ART treatment.

The common AE was infection, followed by bleeding. Our findings are similar to a study on a mature VMMC program in Zimbabwe where infection was the most common AE [16]. However, the occurrence of AEs dropped from 71.7% in 2012 to 3.2% in 2017. This finding agrees with the WHO recommendation that training of VMMC providers prevents occurrence of AEs [7]. The reduction in infection-related AEs from 2,448 of the 2,602 adverse events (94.08%) in 2012 to 129 of the 2,069 adverse events (6.23%) in 2017 is indicative of the improvement in infection prevention and control practices and quality of care. This is inline with a case series analysis of AEs in a large scale VMMC program in Tanzania that demonstrated reduction AEs over time [17]. The relatively higher rates of AEs and low rates of follow-up in our study could be attributed to challenges in documentation at facility level. Close monitoring and documentation of AEs are recommended to help program quality improvement. Our study showed that males circumcised by a Prepex device technique were more likely to develop an AE (OR = 6.98) compared to those circumcised by the surgical technique. This finding corroborates those of another study [16], however, the closure of CIRC MedTech in 2020, the manufacturer of Prepex may mean that the Prepex device will not be in use again. This followed WHO recommendation for tetanus-toxoid vaccine immunization for all males seeking VMMC by Prepex [18].

A majority of the identified AEs in our study were reported in the 15- to 19-year-olds, contrary to findings from another study [16] where clients 10- to 14-years-old contributed most of the AEs, especially those related to infection. The findings of our study could be attributed to the change in policy by the NDoH in 2016 requiring all clients 10–14 years old to be accompanied to the health facility by their parent/guardian on the day of the male circumcision procedure and participate in health education for wound care at home [10]. The high number of infection-related AEs across all age groups indicates need for increased emphasis on interpersonal communication measures at health facilities, clients' health education on wound hygiene while at home, education on avoiding use of herbal medicine and home remedies on the wound and returning for physical review of the wound as per programmatic schedule [19]. The large number of reported AEs with missing data emphasizes the need for VMMC

programs to adequately record, report, manage and monitor all identified AEs to ensure that clients are receiving high quality care and complications are avoided. This is in line with findings from other studies where AEs were found to be poorly recorded and under reported [16, 20]. While reporting AEs, standardized clear classification by severity, type and timing is important to inform the VMMC program of possible gaps in quality of services offered at these facilities.

## Limitations

The following limitations are noted. Firstly, being routine program data, some important variables to inform relationships were missed. For example, the technique used during male circumcision procedure was only recorded as device method or surgical method. The surgical method was not disaggregated further into forceps guided, dorsal slit or sleeve resection methods. Furthermore, only three types of AEs were included (bleeding, infection, and insufficient skin removal) with 31.97% (n = 3,391) missing information on type. Client follow-up was done but this was likely not well-captured in the system. However, the large data set offered a real-world experience thus maximizing generalizability, which often is a challenge in small data sets. Secondly, being routine data, there were several processing challenges due to data capturing, which may lead to misclassification, and subsequent bias. However, manual reviews were conducted, data was verified with source documents thereby improving the quality of the data, giving more representative and generalizable results.

## Conclusion

There was high acceptance of circumcision and low HIV prevalence among the young men. While a majority accepted HIV testing, the proportion of refusals (15%) is still high and requires intensified counselling on benefits of knowing one's HIV status. Both targeted HIV testing and a two-way linkage to care need to be part of comprehensive HIV programs.

There was a relatively low uptake of VMMC services among males 20–34 years old. VMMC programs need to implement innovative demand creation strategies to encourage older males at higher risk of HIV acquisition to get circumcised. This will create an immediate impact in reduction of HIV incidence in these communities. VMMC remains one of the major HIV preventive mechanisms and more demand will help to reach HIV epidemic control. Accurate reporting, management, recording and monitoring of AEs in VMMC programs need to be strengthened to ascertain the quality of services offered at VMMC facilities. Proper training and mentorship are necessary to minimize AEs related to devices.

## Acknowledgments

Kgagamatso Chimelwane and Lynette Stone for assisting with RightMax downloads.

## Author Contributions

**Conceptualization:** Khumbulani Moyo, Nelson Igaba, Skye Grove, Dirk Taljaard, Dino Rech, Arthi Ramkissoon, Catherine Searle, Pappie Majuba, Ian Sanne.

**Data curation:** Motshana Phohole, Caroline Makura, Latisani Mudau.

**Formal analysis:** Constance Wose Kinge, Charles Chasela, Caroline Makura.

**Funding acquisition:** Khumbulani Moyo, Pappie Majuba, Ian Sanne.

**Investigation:** Nelson Igaba, Skye Grove.

**Methodology:** Nelson Igaba, Charles Chasela, Motshana Phohole, Skye Grove, Dirk Taljaard.

**Project administration:** Khumbulani Moyo, Nelson Igaba, Dirk Taljaard, Dino Rech, Arthi Ramkissoon, Catherine Searle, Pappie Majuba, Ian Sanne.

**Validation:** Motshana Phohole.

**Writing – original draft:** Khumbulani Moyo, Charles Chasela.

**Writing – review & editing:** Nelson Igaba, Constance Wose Kinge, Charles Chasela, Motshana Phohole, Skye Grove, Caroline Makura.

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
