## [Decision Letter · Decision Letter 0]

20 May 2021

PONE-D-21-05600

Voluntary Medical Male Circumcision (VMMC) in Selected Provinces in South Africa: Outcomes from a Programmatic Setting

PLOS ONE

Dear Dr. Chasela,

Thank you for submitting your manuscript to PLOS ONE. After careful consideration, we feel that it has merit but does not fully meet PLOS ONE’s publication criteria as it currently stands. Therefore, we invite you to submit a revised version of the manuscript that addresses the points raised during the review process.

Thank you for  submitting this well written manuscript for consideration. The reviewer has come back with some comments for you to address. The reviewer has recommended that you strengthen the discussion and make recommendations for VMMC programmes going forward.

We look forward to receiving your revised manuscript.

Kind regards,

Tendesayi Kufa, MBChB, PhD

Academic Editor

PLOS ONE

Journal Requirements:

[The VMMC program was funded by USAID, award number AID-674-Q-13-00002]

 [NO - The funders had no role in study design, data collection and analysis, decision to publish, or preparation of the manuscript.]

[no].

6. We note you have included a table to which you do not refer in the text of your manuscript. Please ensure that you refer to Table 3 in your text; if accepted, production will need this reference to link the reader to the Table.

Additional Editor Comments:

Thank you for submitting this well written manuscript for consideration. Apologies for the delays in getting back to you with the outcome of the peer review. This is because we battled to get peer reviewers. Now that we managed to get an excellent review, please see comments from the reviewer and some additional ones of my own.

Data sharing= provide details of who and how to contact with data requests

Abstract

Introduction

- the first sentence maybe missing the word "most" before effective

Conclusion

- the authors conclude that targeting young males before 25 will prevent HIV spread and achieve epidemic control. I thought that is what the programme is already doing. Although not intentionally doing so, it has ended up circumcising mostly young males

Introduction

Line 30 - what do the authors mean its exact role in HIV prevention is unknown? That the mechanism by which it prevents HIV is unknown?

Line 45- these data are old. Why are these data still relevant and what additional lessons can they provide to the VMMC programme?

Materials and Methods

Line 54 - were there other partners providing VMMC services in these provinces besides the RTC VMMC programme. If so what was the % of circumcisions contributed by the RTC programme

Line 62- should be USAID not USAIDs

Line 107- why would there have been females in the VMMC database

Line 110- define the different HIV testing outcomes eg Known status

Results

Line 123 - is it possible to compare the provincial number of circumcisions per 100 000 of the population aged 10-49 for each year? Gauteng may account for 46% of the circumcisions but this may not look so impressive after adjusting for the size of the population

Line 127- should this be known HIV positive status

Line 132- 137- is it possible to present AE rates by age?

Table 1

HIV testing offered - should the Not tested (Known HIV+ status). Can this be defined in the methods

Newly diagnosed HIV(%)- shouldn't the denominator for this be the HIV positives?

Table 2

in the row >=40 and column positive , the percentage is listed as 740%. Please check

Discussion

Lines 162- 167- the authors should refer to positivity throughout and not prevalence

Reviewers' comments:

Reviewer's Responses to Questions

**Comments to the Author**

1. Is the manuscript technically sound, and do the data support the conclusions?

Reviewer #1: Yes

2. Has the statistical analysis been performed appropriately and rigorously? 

Reviewer #1: Yes

3. Have the authors made all data underlying the findings in their manuscript fully available?

Reviewer #1: Yes

4. Is the manuscript presented in an intelligible fashion and written in standard English?

Reviewer #1: Yes

5. Review Comments to the Author

Reviewer #1: This is a very relevant manuscript with a large dataset, highlighting experiences from a routine program setting. The discussion section is relatively weak and needs to be strengthened before the manuscript is published. Further details as included in the attachment

6. PLOS authors have the option to publish the peer review history of their article (what does this mean?). If published, this will include your full peer review and any attached files.

Reviewer #1: **Yes: **Augustino Hellar

---

## [Author Response · Author response to Decision Letter 0]

15 Sep 2021

Specific comment to be addressed 

Line 29 – role undetermined – check literature – there is very clear evidence role of VMMC in HIV prevention even beyond the RCTs This has been corrected in the manuscript line 30-32

Line 84 – check sentence for correctness: was VMMC deferred if CD4 count was above or below 350? This has been corrected in the manuscript line 89

Line 93 – It would be ideal to add a reference for MOVE such as:-(https://www.malecircumcision.org/resource/considerations-implementing-models-optimizing-volume-and-efficiency-male-circumcision)-

Reference on MOVE teams has been added in the manuscript line 98 and reference No. 9

Line 107 – why were clients under 10 years included in the database in the first place? Was this a data entry error or was it a violation of the PEPFAR policy for VMMC for HIV prevention which disallows circumcision for clients under 10years?This has been addressed in line 112-113 in the manuscript.

Note: No clients below 10 years were circumcised in this program as they were no eligible.

Line 118-123 – it will be more compelling to include number of clients circumcised by each method of circumcision – i.e. Forceps Guided, Dorsal Slit, sleeve or device methods. If data is available, adverse events should be compared by method of circumcision to be able to compare with WHO recommendations and other studies done elsewhere. This has been addressed in line 227-234 in the manuscript, under limitations

Line 120 – explain what happened to those who consented but did not undergo VMMC. Were they offered services later? This is addressed in the manuscript in lines 139-144.

Line 132 – show the overall AE rate from 2012-2017 in the results paragraph. It is indicated later in Table 3 but it is important to show it early as it is a critical outcome indicator for the paper. Why were the rates calculated out of all VMMCs rather than out of those who returned for follow-up? Please provide an explanation. This has been addressed in table 3 (Line 167) in the manuscript

Table 1 – explain the meaning of “null” in the HIV status in Table 1 

 The “Null” category comprised clients with inconclusive results. This has been updated in the manuscript in table 1 and table 2 (Lines 137 and 151). 

Line 131-138: Consider adding further details on types of AEs in this section, rather than just the three broad categories (infection-related, bleeding-related or insufficient skin removal). Current focus by both WHO and PEPFAR is on notifiable adverse events (NAEs) such as glans injuries and urethral fistulae which have programmatic implications. With such a large database, it is a huge opportunity to describe these critical SAEs (if data is available), rather than just the broad categories and will strengthen this manuscript. This has been addressed in line 224-231 in the manuscript, under limitations

---

## [Editor Report · Decision Letter 1]

9 Nov 2021

PONE-D-21-05600R1Voluntary Medical Male Circumcision (VMMC) in Selected Provinces in South Africa: Outcomes from a Programmatic SettingPLOS ONE

Dear Dr. Chasela,

Thank you for submitting your manuscript to PLOS ONE. After careful consideration, we feel that it has merit but does not fully meet PLOS ONE’s publication criteria as it currently stands. Therefore, we invite you to submit a revised version of the manuscript that addresses the points raised during the review process.

Please address all of the comments raised previously by the reviewer and by the Academic Editor (refer to attached documents). The conclusion (both abstract and main paper) needs reworking. Currently, manuscript states that reaching adolescents has implications for epidemic control yet VMMC programs need to reach 20-29 or even 20-35 year-olds. Implications of VMMC programs continuing to miss the population at greatest risk need to be discussed.

We look forward to receiving your revised manuscript.

Kind regards,

Webster Mavhu

Academic Editor

PLOS ONE

Journal Requirements:

Additional Editor Comments:

This manuscript presents findings from a large set of programmatic data. I have the following comments/suggestions (see also attached document).

1) Authors state that Targeting young men below the age of 25 years will help to prevent HIV spread and accelerate reaching epidemic control. This is incorrect. Actually, programs should reach adult men aged 20-35 years as they are the age group at greatest risk of HIV see for example: Mavhu W, et al. (2021). Innovative demand creation strategies to increase voluntary medical male circumcision uptake: a pragmatic randomised controlled trial in Zimbabwe. BMJ Global Health, 6 (S4): e006141.

2) Related to 1 above, the implications of VMMC programs continuing to miss the age group at greatest risk of HIV needs to be discussed in Discussion.

3) Line 20. There was a reduction in infection-related AEs from 2448 (94.08%) in 2012 to 129 (46.9%) in 2017 of the total AEs reported each year over the five-year period. Please include denominators in both cases (and a p-value).

4) This issue was raised previously by a reviewer - In introduction, you need to cite newer sources e.g.

By December 2019, nearly 27 million adolescent and adult men (≥10 years) had been circumcised and an estimated 340 000 new infections averted in 15 VMMC priority countries, including 260 000 infections among males and 75 000 among females (due to reduced secondary transmission from males)

WHO. Preventing HIV through safe voluntary medical male circumcision for adolescent boys and men in generalized HIV epidemics: recommendations and key considerations. Geneva: WHO, 2020. UNAIDS/WHO. Voluntary Medical Male Circumcision: Steady progress in the scaleup of VMMC as an HIV prevention intervention in 15 eastern and southern African countries before the SARS-CoV2 pandemic. Geneva: UNAIDS and WHO, 2021.

5) Consent procedures (lines 70-77) describe consent procedures for everyone 11 and above but not 10 year-olds?

6) The conclusion needs strengthening - both in Abstract and Main paper.

7) Line 252 - While, most accepted HIV testing, the number of refusals is still high. If 99.6% took up VMMC then only 0.45 refused?

8) The manuscript needs thorough editing/proof reading (see attached edits).
---

## [Author Response · Author response to Decision Letter 1]

3 Dec 2021

Specific comment to be addressed Responses 

Include denominator for both cases This has been addressed in line 19

Programs should reach adult men aged 20-35 years as they are the age group at greatest risk of HIV This has been addressed in line 23-25

Agree with 1 reviewer that you need to cite newer sources in the introduction Newer sources have been cited in the introduction in line 31, 35, 38, 40

Consent procedures (lines 70-77) describe consent procedures for everyone 11 and above but not 10-year-olds? This has been addressed in line 79

Which program – the one under review or the referenced one? If the latter, include reference. If this one, see earlier comment. The age group most missed by VMMC programs (20-35) is the most at risk.

 This has been addressed in line 185-188

Include references This has been addressed in line 189

Line 252 - While, most accepted HIV testing, the number of refusals is still high. If 99.6% took up VMMC then only 0.45 refused? This has been addressed in Table 2. 

A total of 1,001,088 were offered an HIV test and 71.1% were tested, 15% declined to test

Authors state that Targeting young men below the age of 25 years will help to prevent HIV spread and accelerate reaching epidemic control. This is incorrect. Actually, programs should reach adult men aged 20-35 years as they are the age group at greatest risk of HIV see for example: Mavhu W, et al. (2021). Innovative demand creation strategies to increase voluntary medical male circumcision uptake: a pragmatic randomised controlled trial in Zimbabwe. BMJ Global Health, 6 (S4): e006141 This has been addressed in line 38-39, 187-188

Related to above, the implications of VMMC programs continuing to miss the age group at greatest risk of HIV needs to be discussed in Discussion. This has been addressed in line 185-198

The conclusion needs strengthening - both in Abstract and Main paper. This has been addressed in line 21-26 and 265-276 

The manuscript needs thorough editing/proof reading (see attached edits). This has been addressed from line 1-352

---

## [Editor Report · Decision Letter 2]

23 Dec 2021

PONE-D-21-05600R2

Voluntary Medical Male Circumcision  in Selected Provinces in South Africa: Outcomes from a Programmatic Setting

PLOS ONE

Dear Dr. Chasela,

Thank you for submitting your manuscript to PLOS ONE. After careful consideration, we feel that it has merit but does not fully meet PLOS ONE’s publication criteria as it currently stands. Therefore, we invite you to submit a revised version of the manuscript that addresses the points raised during the review process.

It is now 4 years since these data were collected. To what extent are the findings and conclusions still applicable/relevant? If still relevant, the paper could say in a couple of places, how the data compare to what is currently obtaining within the VMMC program. 

Authors mention a device (probably PrePex). Recommendations related to this device may no longer apply as the device has now been taken off the market and may never be reintroduced. Perhaps recommendations could focus on VMMC devices in general? 

We look forward to receiving your revised manuscript.

Kind regards,

Webster Mavhu

Academic Editor

PLOS ONE

Journal Requirements:

Additional Editor Comments (if provided):

It is now 4 years since these data were collected. To what extent are the findings and conclusions still applicable/relevant?

If still relevant, the paper could say in a couple of places, how the data compare to what is currently obtaining within the VMMC program.

Authors mention a device (probably PrePex). Recommendations related to this device may no longer apply as the device has now been taken off the market and may never be reintroduced. Perhaps recommendations could focus on VMMC devices in general?

See additional suggestions in attached document.
---

## [Author Response · Author response to Decision Letter 2]

6 Feb 2022

We have addressed the relevance of the paper in the discussion section and provided the citation as requested.

---

## [Editor Report · Decision Letter 3]

1 Apr 2022

PONE-D-21-05600R3Voluntary Medical Male Circumcision  in Selected Provinces in South Africa: Outcomes from a Programmatic SettingPLOS ONE

Dear Dr. Chasela,

Thank you for submitting your manuscript to PLOS ONE. See a few suggestions in attached.

We look forward to receiving your revised manuscript.

Kind regards,

Webster Mavhu

Academic Editor

PLOS ONE

Journal Requirements:

Additional Editor Comments (if provided):

Find a few suggestions in the attached.
---

## [Author Response · Author response to Decision Letter 3]

20 May 2022

We have revised the references as well as changed all decimal points to 2.

---

## [Editor Report · Decision Letter 4]

14 Jun 2022

Voluntary Medical Male Circumcision  in Selected Provinces in South Africa: Outcomes from a Programmatic Setting

PONE-D-21-05600R4

Dear Dr. Chasela,

We’re pleased to inform you that your manuscript has been judged scientifically suitable for publication and will be formally accepted for publication once it meets all outstanding technical requirements.

Kind regards,

Webster Mavhu

Academic Editor

PLOS ONE
---

## [Editor Report · Acceptance letter]

12 Sep 2022

PONE-D-21-05600R4 

Voluntary Medical Male Circumcision in Selected Provinces in South Africa: Outcomes from a Programmatic Setting 

Dear Dr. Chasela:

I'm pleased to inform you that your manuscript has been deemed suitable for publication in PLOS ONE. Congratulations! Your manuscript is now with our production department. 

Kind regards, 

on behalf of

Dr. Webster Mavhu 

Academic Editor

PLOS ONE